# Establishment and Verification of a Novel Gene Signature Connecting Hypoxia and Lactylation for Predicting Prognosis and Immunotherapy of Pancreatic Ductal Adenocarcinoma Patients by Integrating Multi-Machine Learning and Single-Cell Analysis

**DOI:** 10.3390/ijms252011143

**Published:** 2024-10-17

**Authors:** Ying Zheng, Yang Yang, Qunli Xiong, Yifei Ma, Qing Zhu

**Affiliations:** Division of Abdominal Tumor Multimodality Treatment, Cancer Center, West China Hospital, Sichuan University, No. 37 Guoxue Alley, Chengdu 610041, China; ying_zheng1@163.com (Y.Z.);

**Keywords:** histone lactylation, hypoxia, lactic acid, immunotherapy, pancreatic ductal adenocarcinoma, chemotherapy

## Abstract

Pancreatic ductal adenocarcinoma (PDAC) has earned a notorious reputation as one of the most formidable and deadliest malignant tumors. Within the tumor microenvironment, cancer cells have acquired the capability to maintain incessant expansion and increased proliferation in response to hypoxia via metabolic reconfiguration, leading to elevated levels of lactate within the tumor surroundings. However, there have been limited studies specifically investigating the association between hypoxia and lactic acid metabolism-related lactylation in PDAC. In this study, multiple machine learning approaches, including LASSO regression analysis, XGBoost, and Random Forest, were employed to identify hub genes and construct a prognostic risk signature. The implementation of the CERES score and single-cell analysis was used to discern a prospective therapeutic target for the management of PDAC. CCK8 assay, colony formation assays, transwell, and wound-healing assays were used to explore both the proliferation and migration of PDAC cells affected by *CENPA*. In conclusion, we discovered two distinct subtypes characterized by their unique hypoxia and lactylation profiles and developed a risk score to evaluate prognosis, as well as response to immunotherapy and chemotherapy, in PDAC patients. Furthermore, we indicated that *CENPA* may serve as a promising therapeutic target for PDAC.

## 1. Introduction

Pancreatic ductal adenocarcinoma (PDAC) is recognized as the deadliest malignant tumor, ranking seventh among the leading causes of cancer-related fatalities worldwide and sixth in China [1,2]. With its high invasiveness, rapid disease progression, and the lack of early diagnostic methods for diagnosis, pancreatic cancer exhibits the bleakest prognosis compared to other cancer types, with a 5-year overall survival of only 10% [3]. Over the past few years, there has been a gradual rise in the incidence of PDAC, and it is predicted to emerge as the second-most prevalent cause of cancer-related mortality in the United States [2]. Surgical resection remains the most effective treatment for patients diagnosed at an early stage. However, recent advances in chemotherapy and immunotherapy have yielded limited improvements in PDAC outcomes [4], primarily due to the unique tumor microenvironment characterized by hypoxia and metabolic reprogramming. Nonetheless, strategies targeting the remodulation of the PDAC tumor microenvironment have shown promising results.

The microenvironment of PDAC is commonly characterized by severe hypoxia, compared to adjacent normal pancreas [5]. The specific tumor microenvironment (TME) of PDAC can arise from several factors, including the presence of abundant fibrotic stroma, rapid proliferation of cancer cells, and reduced angiogenesis [6]. Hypoxia stimulates the expression of various angiogenic factors, thereby facilitating tumor angiogenesis and creating a microenvironment conducive to tumor cell proliferation [7]. Concurrently, tumor cells undergo a metabolic shift from oxidative phosphorylation to glycolysis under hypoxic conditions, resulting in enhanced ATP production to support their growth and progression. Furthermore, the hypoxic microenvironment induces remodeling of the extracellular matrix, altering its biochemical characteristics to promote tumor proliferation and metastatic potential [8]. Moreover, hypoxia promotes epithelial–mesenchymal transition (EMT) in tumor cells. Research has demonstrated that hypoxia induces the interaction between *TWIST* and Ring1B, as well as *EZH2*, thereby facilitating the proliferation and metastasis of pancreatic cancer in mouse xenograft models [9]. Recently, hypoxia has emerged as a significant factor contributing to immune evasion and undermining the efficacy of immunotherapy [10]. In response to hypoxia, pancreatic cancer cells activate hypoxia-induced factors (HIFs), which play a pivotal role in the regulation of genes involved in angiogenesis, migration, and invasion of pancreatic cancer cells. Notably, *HIF-1* is a key player in balancing oxidative and glycolytic metabolism [11]. In hypoxic environments, cancer cells switch from oxidative to glycolytic metabolism to sustain ATP production and protect against cytotoxic reactive oxygen species (ROS) while concurrently leading to lactate accumulation. The elevated lactate levels within the PDAC tumor microenvironment contribute to the establishment of an immunosuppressive microenvironment.

The conventional belief regarding lactate as a metabolic waste has been challenged in recent years. Emerging evidence suggests that lactate serves as a fuel for mitochondrial metabolism and plays a critical role in tumor progression and influencing the function of immune cells, ultimately contributing to the formation of an immunosuppressive environment within the tumor microenvironment (TME) [12]. The dense fibrous tissue in the pancreatic cancer tumor microenvironment not only compromises vascular integrity, inducing hypoxia, but also increases lactate levels in the tumor microenvironment by activating the *PI3K/Akt* pathway to promote anaerobic glycolysis [13]. Studies suggests that elevated lactate levels may be the cause of radiotherapy-induced myeloid-derived suppressor cells promoting an immunosuppressive phenotype in pancreatic cancer patients. Simultaneously, the regulation of this phenomenon by *HIF-1α* through the *(GPR81)/mTOR/HIF-1α/STAT3* pathway is crucial [14]. Elevated lactate levels in the tumor microenvironment are not only associated with the energy metabolism of a tumor but may also trigger epigenetic alterations within the tumor. Additionally, Di Zhang et al. identified a novel protein modification known as histone lactylation, providing us with a different insight into the relationship between hypoxia and posttranslational modification. Their research revealed that hypoxia induces lactate production through glycolysis, which subsequently triggers lactate-mediated histone lactylation at lysine residues. Remarkably, the level of histone lactylation is closely associated with the intracellular lactate level [9]. These biological processes have implications in various pathophysiological conditions, including cancer progression. Research has shown that lactylation of histone H3 at lysine 18 (*H3K18la*) drives the expression of the key transcription factor *YBX1*, leading to cisplatin resistance in bladder cancer [15]. Similarly, in colorectal cancer, the metabolic byproduct lactate promotes *H3K18* lactylation, which, in turn, induces *RUBCN1*-mediated resistance to bevacizumab in tumor cells [16]. Additionally, another study found that lactylation of *HIF-1α* induces the expression of *KIAA1199*, thereby facilitating angiogenesis in prostate cancer [17]. Despite numerous studies indicating that lactylation influences tumor progression, metastasis, and drug resistance through mechanisms such as metabolic reprogramming and epigenetic modifications, research on lactylation in tumor advancement remains in its early stages. There is still a critical need for extensive investigation to explore its biological significance further [18]. However, the remodeling of the TME induced by hypoxia and histone lactylation in PDAC remains unexplored.

In our study, we identified 33 geneslinked to hypoxia and lactylation, which are tied to the prognosis ofpancreatic ductal adenocarcinoma (PDAC). Given that gene mutation frequency is significantly associated with tumor development, and genes with high mutation rates may serve as key drivers of tumorigenesis, we also analyzed the mutation frequency of these genes in pancreatic cancer. Based on the expression of these genes, we categorized PDAC patients into two subtypes exhibiting distinct survival differences. Next, we investigated the mutation frequency, biological disparities, and clinical characteristics between these subtypes. Subsequently, leveraging the prognostic value of these hypoxia- and lactylation-related genes, we developed a risk score that demonstrates some predictive power for survival, immunotherapy, and chemotherapy outcomes of PDAC patients. Importantly, we discovered a significant signature gene, *CENPA*, which may serve as a promising therapeutic target for PDAC.

## 2. Results

### 2.1. Identification of Prognostic Hypoxia- and Lactylation-Related Genes and Their Genetic Variations in PDAC

In this study, a combination of gene transcription profiles of PDAC patients from the TCGA and GETx databases was used to identify differentially expressed genes (DEGs) in PDAC. With the utilization of R package “limma”(version 3.58.1) (log2 (FC)| > 1.5, *p* < 0.05), we obtained a total of 2333 differentially expressed genes(DEGs). According to the acknowledgement that lactylation modification is closely related to glycolysis, lactate metabolism, and hypoxia, we identified four lactylation-related pathways and two hypoxia-related gene sets in the GSEA database and extracted 764 hypoxia- and lactylation-related genes (HALRGs). By taking the intersection of DEGs and hypoxia- and lactylation-related genes, we obtained 87 differentially expressed hypoxia- and lactylation-associated genes (Figure 1A). The univariate Cox regression analysis was performed later in a TCGA and GSE183795 combined mRNA expression cohort to identify 31 prognostic hypoxia- and lactylation-related genes, and a forest diagram was plotted based on the results of the above univariate Cox regression analysis (Figure 1B). A total of 31 prognostic HALRGs showed prominent prognostic capacity, and Kaplan–Meier survival curves of *PLOD1*, *LDHA*, *PIGA*, and *CENPA* are shown in Figure 1D and Appendix A.

The occurrences of copy number variations (CNVs) and somatic mutations of 31 prognostic genes are summarized in Figure 1C. Within the dataset of 173 tumor samples, the waterfall plot in Appendix A reveals that these 31 prognostic genes underwent mutations at a frequency of 4.05%, with *P4HA1* displaying the highest mutation rate. *TIMM50* exhibits the highest frequency of copy number gain, while *ERRFI1* has the most significant CNV loss, as shown in Appendix A. To study the biological network integration for gene prioritization and predicting the gene function of those prognostic genes, GeneMANIA prediction server (http://www.genemania.org, accessed on 3 January 2024) was used in this study [19]. In the circle diagram, genes in the inner circle are prognosis-related genes, and outer circle genes are those that the sets predict are functionally similar or have shared properties with the initial prognostic genes. The interaction and function between these genes are visualized in Figure 1C.

### 2.2. Pan-Cancer Analysis of the Prognostic HALRGs

Since we already identified the prognostic values of 31 HALARGs in PDAC patients, to deepen our understanding of the biological functions and clinical values of those prognostic genes, we conducted a pan-cancer analysis by using Gene Set Cancer Analysis (GSCA). We aim to identify the commonalities and specificities of these genes in tumor development through pan-cancer analysis, thereby enhancing the broad applicability and generalizability of our research. First of all, GSCA estimates the association between GSVA scores based on the 31 genes and survival (includes OS, PFS, DSS, and DFI) in 33 cancer types. Significant survival differences between low and high GSVA score groups across multiple cancer types can be observed in Figure 2A, including some common cancer types like lung adenocarcinoma (LUAD), bladder urothelial carcinoma (BLCA), kidney renal clear cell carcinoma (KIRC), mesothelioma (MESO), head and neck squamous cell carcinoma (HNSC), and so on, particularly among patients with pancreatic adenocarcinoma. The bubble plot visually demonstrates these disparities, indicating that the higher GSVA score group has a poor prognosis. These results verified our previous results from the univariate Cox regression analysis and suggest that it is necessary to investigate the biological value of those genes in PDAC patients. To further analyze the biological mechanisms behind it, we investigated the association between GSVA score and activity of cancer-related pathways in pan-cancer. We found that the cell cycle pathway and apoptosis pathway were activated in most cancers, including LUAD, LIHC, HNSC, COAD, STAD, and so on. Interestingly, we found that the EMT pathway was activated in pancreatic cancer, indicating that patients with a higher expression level of these prognostic genes may experience a higher risk of tumor progression and metastasis (Figure 2B). Subsequently, we examined the disparities in prognostic gene expression and pathway activity groups, namely activation and inhibition, classified based on the median pathway scores. The heatmap illustrates the percentage of cancers across 33 cancer types wherein a gene has a significant effect (FDR ≤ 0.05) on a particular pathway. Based on the heatmap analysis, we posit that the mRNA expression levels of several prognostic genes potentially activate the cell cycle, EMT, and apoptosis pathway across various cancer types (Appendix A).

Next, we used GSCA to analyze the correlation between prognostic gene expression and sensitivity of GDSC and CTRP drugs (top 30) in pan-cancer (Figure 2C,D). The bubble plot demonstrated that the mRNA expression level of *SLC2A1*, *PLOD1*, *PKP2*, *ERRFI1*, *ALDH1B1*, and *TIPARP* is positively correlated with the sensitivity of multiple GDSC drugs. While they have a native correlation with the sensitivity of some GDSC drugs, including trametinib, a commonly used chemotherapeutic agent for abdominal tumor, at the same time, the mRNA expression levels of *PPIA*, *SLC25A42*, *KDM3A*, *CHEK2*, and *NUP98* have the reverse outcome.

### 2.3. Unsupervised Clustering Analysis Identified Two Prognostic HALRAGs-Related Subtypes

Based on the gene expression of 31 HALRGs, we performed a consensus clustering analysis to classify the PDAC samples (TCGA and GSE183795 combined expression cohorts, *n* = 318). As shown in Figure 3A, we successfully separated all PDAC samples into two distinct clusters, and the optimal clustering variable was 2. The result was later confirmed by the PCA (Figure 3B). We employed the “survival” R package (version 3.5.7) to conduct a survival analysis and assess the overall survival (OS) time of the two subtypes. The results indicate that cluster B exhibited a superior prognosis compared to cluster A (Figure 3C). In addition, we extracted the differentially expressed genes between the two subtypes by using the “limma” R package(version 3.58.1) for further analysis (log2 (FC)| > 1.5, *p* < 0.05).

### 2.4. Biological Function Difference between the Two Subtypes

To investigate the underlying biological activity between the two hypoxia- and lactylation-related subtypes, we employed PROGENy (Pathway RespOnsive GENes for activity inference) to assess the activation levels of the cell signaling pathways in both subtypes. Our findings demonstrate differential activation of multiple cell-signaling pathways, including EGFR, hypoxia, JAK-STAT, MAPK, NF-κB, P53, PI3K, and VEGF, among others (Figure 3D). Cluster A exhibited significant activation of oncogenic pathways, including EGFR, MAPK, TGF-β, P53, VEGF, WNT, and hypoxia. This result inspires us to further investigate the biological differences of the two clusters. Other existing variations in the two subtypes may offer a novel insight into their distinct prognostic outcomes.

Next, we performed GO and KEGG functional enrichment analysis of the differentially expressed genes between the two clusters. KEGG analysis revealed that the DEGs were primarily enriched in the cell cycle, HIF-1 signaling pathway, focal adhesion, and Hippo signaling pathway, indicating differences in the tumor microenvironment (TME) between the two groups (Figure 3E). GO analysis showed that the DEGs were associated with wound healing, cell growth, response to hypoxia, and response to oxidative stress (Figure 3F).

### 2.5. Construction and Validation of the Hypoxia- and Lactylation-Related Prognostic Model in PDAC

Based on the previous analysis, we acknowledged the significance of these hypoxia- and lactylation-related genes as prognostic indicators in PDAC patients, indicating the feasibility of constructing prognostic models based on them. We employed three machine learning algorithms (LASSO, XGBoost, and random forest) to screen hub genes for building a prognostic model (Appendix A). Eight genes (*LDHA*, *PIGA*, *PLOD1*, *POMT1*, and *MPC1*) were meticulously selected through the intersection of genes identified by diverse machine learning algorithms, forming the basis for constructing the subsequent model. Utilizing the expression levels and coefficients derived from LASSO regression analysis for each gene, we computed a risk score for every sample (Figure 4A–C) (Risk score = β*_LDHA_* × Expression*_LDHA_* + β*_PIGA_* × Expression*_PIGA_* + β*_PLOD1_* × Expression*_PLOD1_* + β*_POMT1_* × Expression*_POMT1_* + β*_MPC1_* × Expression*_MPC1_* + β*_KDM3A_* × Expression*_KDM3A_* + β*_SLC25A4_* × Expression*_SLC25A4_* + β*_CENPA_* × Expression*_CENPA_*, β = coefficient of each gene). All the samples of the training cohort were divided into low- and high-risk groups based on the median risk score. The survival curves revealed that the group with higher risk scores experienced a poor prognosis (Figure 4G). In the training cohort, AUC values at 1, 3, and 5 years are 0.674, 0.742, and 0.731, respectively, indicating that the risk model has potential predictive power for PDAC patients (Figure 4H). Another ROC curve suggests that, compared to other clinical features, the risk score has the highest AUC value (Figure 4I). The expression profile of the eight hub genes in the training cohort is shown in the heatmap (Figure 4D). Figure 4E,F illustrate the presentation of the risk score distribution and survival status.

Furthermore, for a more comprehensive assessment of the prognostic model’s reliability and validity, we utilized GSE62452, GSE78299, and GSE85916 as three external independent test cohorts in the subsequent analysis. All samples were divided into low- and high-risk groups based on the prognostic model. Similarly, the survival analysis demonstrated a better overall survival (OS) in low-risk patients compared to high-risk patients (Figure 5A–C). The AUC values at 1, 3, and 5 years for the GSE62452 test set are 0.619, 0.760, and 0.903, respectively (Figure 5D). In the GSE78229 test set, the AUC values at 1, 3, and 5 years are 0.650, 0.760, and 0.876, respectively (Figure 5E). In the GSE85916 test set, the AUC values at 1, 3, and 5 years are 0.632, 0.583, and 0.632 (Figure 5F). The ROC curves of the three test sets confirmed some prognostic predictive value of the prognostic model for PDAC patients.

### 2.6. Development of the Predictive Nomogram and Mutation between Risk Groups

The univariate and multivariate Cox regression analyses both illustrated the reliability of the risk score as an independent prognostic factor compared to other clinical characteristics (Figure 5I,J). Subsequently, we attempted to incorporate clinical parameters to construct a nomogram, which serves as a valuable tool for predicting the 1-, 3-, and 5-year overall survival (OS) of PDAC patients (Figure 5G). The calibration curve is depicted in Figure 5H. Next, a waterfall plot was employed to visually represent gene mutations in both the low- and high-risk groups. It is worth noting that the high-risk groups exhibit a substantially higher mutation burden. Specifically, the mutation frequency of KRAS and TP53 is significantly elevated in the high-risk group (Figure 4I,J).

### 2.7. Immunologic Landscape and Functional Analysis of the Signature

According to the previous literature, histone lactylation and hypoxia have an essential impact on metabolic reprogramming of tumors and thus affect the immunotherapy response of multiple cancer types. We firstly use multiple immune algorithms to assess the correlation between risk scores and immune cells abundance (Figure 6A). The founding suggested a negative correlation between risk scores and the majority of infiltrating immune cells, indicating that patients with higher risk scores may have a more severe immunosuppressive microenvironment. Furthermore, we also investigated the correlation between signature genes and immune cells (Appendix A). Of note, *CENPA* is strongly positive related to M0 macrophages and dendritic cells activated. Moreover, we obtained several immune escape-related genes from the previous literature to assess their association with the risk score. Figure 6C demonstrates a strong positive correlation between the risk score calculated by our prognostic model and the majority of immune escape-related genes. All these findings suggest that the prognostic module model may serve as a valuable tool for predicting the immunotherapy response of PDAC patients.

### 2.8. Prediction of Immunotherapy Response and Chemotherapy Sensitivity Analysis

Drawing from the preceding analysis, we have determined an underlying link between the risk scores and tumor immunity. Therefore, we used Tumor Immune Dysfunction and Exclusion (TIDE) algorithms to explore the immunotherapy response of PDAC patients in low- and high-risk groups. Our fundings suggested that differences in immunotherapy responses do exist in patients from the two groups, with the low-risk group experiencing a higher response rate to immunotherapy (Figure 6B). Then, we used the “oncoPredict” R package (version 0.2) to predict 198 clinical drug responses in TCGA-PAAD patients divided into low- and high-risk groups. Figure 6D revealed a significant finding where patients in the low-risk group demonstrated increased sensitivity to a wide range of commonly used anti-tumor medications, including Sorafenib, Oxaliplatin, Dabrafenib, Irinotecan, and others. Those findings provided evidence that our prognostic model would probably be able to function as an effective tool in predicting the responsiveness of chemotherapy drugs for PDAC patients.

### 2.9. CERES Score and Character of the Signature Genes by scRNA-Seq Analysis

To identify the most essential gene from all signature genes, CERES scores were calculated among 43 PDAC cell lines, following a previous study. We found that the *CENPA* gene owns the lowest score, which means this gene might play a more vital role for PDAC cells (Figure 7A). To comprehensively characterize the signature genes, we conducted a single-cell analysis using the GSE197177 dataset, which comprises four primary PDAC samples, three liver metastasis samples, and one normal sample. As shown in Figure 7B,C, the cell identity of each subtype was annotated by using classical markers of cell subset definition. The average expression level of *CENPA* in each cell type is visualized in Figure 7C. In primary pancreatic cancer single-cell samples, *CENPA* is highly expressed in the ductal cell population, while it is expressed at a lower level in normal samples and metastatic samples, implying that *CENPA* could serve as a potential therapeutic target for pancreatic cancer. Considering the predictive capability demonstrated by our prognostic model, we attempted to conduct more research at the single-cell level. First, we calculated scores for the single-cell samples of both primary and metastatic pancreatic cancer based on the expression of the featured genes using “AddModuleScore” in the “Seurat” R package (version 5.0.1) (Appendix A). To further investigate the biological characteristic differences underlying the low- and high-scoring groups, we explored the complex communications between nine annotated cell types. Compared to the high-scoring group, the low-scoring group showed an increase in the quantity of communication among endothelial cells, fibroblasts, and myeloid cells. Additionally, in the high-scoring group, the number of interactions between endocrine cells and several other cell types was intensified, while the communication intensity between T cells and myeloid cells, as well as acinar cells and ductal cells, weakened (Figure 7D,F). We also identified and prioritized key signaling pathways that showed significant differences between the low- and high-scoring groups. Cells in the high-scoring group activated most of the signaling pathways associated with tumor initiation and progression, including TGF-β, CCL, PTN, FGF, VEGF, and EGFR (Figure 7E).

### 2.10. Knockdown of CENPA Hampers the Proliferation and Migratory Potential of PDAC Cells

TCGA and GETx data of PDAC patients indicated that *CENPA* is highly expressed in the tumor sample (Figure 8B). The immunohistochemistry result obtained from the HPA database validates that the protein expression level of *CENPA* is higher in the PDAC tissue (Figure 8A). To evaluate the prognostic value of *CENPA*, we found that PDAC patients with a high expression level of *CENPA* tend to experience poorer prognosis (Figure 8C). Additionally, we verified the mRNA expression level of *CENPA* in serval PDAC cell lines. A relatively high expression level of *CENPA* was found in BXPC-3, CAPAN-1, CAPAN-2, CFPAC-1, MIA PaCa-2, PANC-1, and SW1990 compared to hTERT-HPDE (Figure 8D). To delve deeper into the influence of *CENPA* on the growth and migration of pancreatic cancer, we developed PANC-1 and MIA Paca-2 cell lines with *CENPA* knockdown (Figure 8E). After conducting CCK-8 assays and clone formation assays, we experimentally demonstrated that the knockdown of *CENPA* effectively hampers the growth and proliferation of PANC-1 and MIA Paca-2 cells (Figure 8F,G). In order to assess the impact of *CENPA* on cellular metastatic potential, we conducted transwell migration and wound-healing assays. Figure 8H shows that the *CENPA* knockdown cells exhibited significantly wider scratch gaps compared to *CENPA* NC cells. Likewise, the transwell migration assay revealed that knockdown of the *CENPA* gene inhibited the migration of PANC-1 and MIA Paca-2 cells (Figure 8I). In light of the aforementioned results, we posit that the gene *CENPA* holds promise as a potential therapeutic target for PDAC.

### 2.11. Correlation Analysis between Drug Sensitivity and the Expression of CENPA and Molecular Docking

Based on our experimental results, we observed an elevated expression of *CENPA* in tumor tissues, which showed a negative correlation with patient prognosis. We further aim to investigate whether there are potential anti-tumor drugs associated with *CENPA*. First, we investigated small molecule anti-tumor drugs related to the expression levels of the *CENPA* gene in multiple pancreatic cancer datasets through the BEST website (Figure 9A). This approach allowed us to identify drugs that may specifically target the pathways influenced by *CENPA*. We selected two anti-tumor drugs from two different databases that are positively correlated with high *CENPA* expression, including betulinic acid and GSK2126458. Then, we conducted molecular docking to investigate the binding affinity of these two drugs with *CENPA*. This computational approach is essential for predicting how well these drugs can interact with the target protein, which is a crucial step in drug development. The results showed that both drugs exhibit a strong affinity for *CENPA*, betulinic acid (−8.1 kcal/mol), and GSK2126458 (−8.6 kcal/mol) (Figure 9B). These values indicate favorable binding interactions, suggesting that both drugs could potentially inhibit *CENPA* activity or disrupt its function in tumor cells. Overall, our findings indicate that *CENPA* is not only a marker of poor prognosis in pancreatic cancer but also a potential therapeutic target. Future studies will be needed to validate the efficacy of betulinic acid and GSK2126458 in preclinical models, as well as to explore the underlying mechanisms by which *CENPA* contributes to tumorigenesis and drug resistance.

## 3. Discussion

While previous studies have characterized lactate as metabolic waste, recent research has revealed its diverse roles in both metabolic and nonmetabolic functions in various diseases, including cancer. Di Zhang et al. recently identified lactylation as a novel epigenetic modification induced by lactate. Several studies have suggested that lactylation promotes cancer progression and impacts tumor immunosuppression by regulating metabolic reprogramming [8]. Moreover, a recent study on ocular melanoma revealed a significant correlation between elevated amounts of histone lactylation and unfavorable prognosis in patients. Additionally, the inhibition of lactylation was found to suppress the proliferation of ocular melanoma cells in vitro [20]. Furthermore, several studies have suggested an intimate link between lactylation and the process of macrophage polarization and activation [21]. Serval studies have focused on targeting lactate or lactylation as a therapeutic opportunity.

Cancer cells often adjust their metabolism in response to hypoxia by increasing glycolysis and decreasing oxidative phosphorylation, which increases lactate generation. It has been reported that hypoxia induces intracellular lactate production and raises histone Kla levels, while not affecting histone Kac levels, in MCF-7 cells [8]. In pancreatic cancer, the hypoxia microenvironment was already recognized as a main factor for enhancing tumor progression and resistance to immunotherapy. The presence of hypoxia areas within PDAC is also considered one of the independent factors with the prognostic value [22].

In summary, considering that hypoxia leads to lactate accumulation, and lactylation is closely associated with the lactate levels, we hypothesize hypoxia and lactylation may be linked in the biological metabolism of pancreatic cancer progression, resulting in the poor prognosis of PDAC patients. However, the role of lactylation and hypoxia in the prognosis of PDAC patients remains unknown. In this study, we used mRNA expression data and clinical information collected from the TCGA and GEO databases to explore the possibility that lactylation and hypoxia might elucidate the prognosis of PDAC patients. The classification of cancer subtypes based on the expression of prognostic genes related to hypoxia and lactylation demonstrated a notable disparity in survival rates between the two subtypes. Cluster A, characterized by a higher expression level of HALRGs, is associated with a poor prognosis in PDAC patients. These findings align with previous studies reported in the literature. Chiou et al. reported that *BLMP1* regulates hypoxia-associated gene expression, promoting metastasis in PDAC and leading to a poor prognosis in patients [23]. PROGENy analysis further revealed the activation of multiple oncogenic pathways in Cluster A, including the hypoxia- and MAPK-related pathways. GSVA analysis further confirmed the significant variations in biological characteristics between Cluster A and Cluster B. To further understand the biological function differences between these two clusters, we also conducted GO and KEGG enrichment analyses in the DEGs of these two clusters; the HIF-1 signaling pathway and ECM–receptor interaction were enriched. More recently, HIFs have been regarded as major players in cancer immune evasion [24]. Noman et al. stated that HIF-1α plays a direct role in the upregulation of *PD-L1* in diverse tumor cells by binding to the hypoxia response element (HRE) within the *PD-L1* gene promoter. This indicates that the hypoxic tumor microenvironment facilitates the development of immunosuppression [25]. Given the recognition that lactate and acidification of the TME are also key promoters of immune escape and modulating the function of immune cells [26], we analyzed the immune infiltration of these two clusters. Unexpectedly, a significant difference was found between these two clusters, especially in activated B cells and activated CD8 T cells.

A primary cellular response to severe hypoxia involves the metabolic shift from oxidative phosphorylation to glycolysis, resulting in the production of lactate. Pancreatic cells are known to possess an enhanced glycolytic capacity due to their distinct tumor microenvironment (TME), which triggers the robust activation of enzymes and transporters associated with lactic acid production [27]. In order to evaluate the prognostic significance of those HALRGs more accurately, we used various machine learning algorithms to screen hub genes and developed a risk signature based on the expression of signature genes. Our prognostic model has demonstrated some predictive power through validation in three independent GEO datasets. In recent years, immunotherapy has established itself as a fourth cornerstone in the treatment of various solid tumors. Despite numerous preclinical and clinical studies indicating the potential of immunotherapeutic approaches for patients with pancreatic ductal adenocarcinoma (PDAC), their efficacy has not yet met expectations [27]. Whether PDAC patients will benefit from immunotherapy still being a hot topic worth exploring, given our observation of a negative correlation between the risk score and the majority of immune-infiltrating cells in various immune cell algorithms, we delved deeper into disparities in immune therapy responses between high- and low-risk groups. Our findings revealed a significantly diminished immune therapy response in patients from the high-risk group compared to their low-risk counterparts. The mechanism behind how a genetic circuit enables cancer cells to evade destruction by cytotoxic T lymphocytes (CTLs) remains poorly known. Recently, Lawson et al. firstly identified 182 genes that were involved in the process of increasing either the sensitivity or the resistance of cancer cells to CTL-mediated toxicity by performing a genome-wide CRISPR screening across multiple mouse cancer modules [28]. To investigate the reasons behind this difference, we analyzed the correlation between the risk score and genes associated with immune escape. Our analysis revealed a positive correlation between the risk score and most immune escape-related genes. This finding aligns with previous studies suggesting that lactate significantly inhibits immune cell function through lactylation in the tumor microenvironment (TME) [29]. Additionally, we observed that high-risk patients exhibited reduced infiltration of tumor-killing immune cells in their immune microenvironment. Thus, we speculate that hypoxia and lactylation may lead to decreased immune cell infiltration, subsequently promoting immune escape. While our study highlights these intriguing associations, we acknowledge that the mechanistic links between risk scores and immune evasion require further investigation. Chemotherapy is still regarded as the primary treatment for PDAC patients who have missed the chance for surgical intervention. Some studies have shown that hypoxia in TME induce resistance to cytotoxic chemotherapy in cancer cells. Xianbin Zhang et al. demonstrated that LW6, a chemical inhibitor of HIF-1α enhances drug sensitivity of gemcitabine in PC cells through suppression of autophagy [30]. Metabolic reprogramming of glycolysis has considered one of the most prominent features of PC cells in response to hypoxia, resulting in the accumulation of lactate. In our study, we discovered a difference of chemotherapy sensitivity between the low- and high-risk groups in a PDAC patient cohort, which is in accordance with previous studies.

*CENPA*, identified as a centromere-specific histone variant of histone H3, belongs to one of the four primary classes of histones (H2A, H2B, H3, and H4). It induces a less compact DNA-binding structure, thereby enhancing the dynamics of DNA ends within the nucleosome core particle [31]. The distinctive characteristic of *CENPA* allows it to play a dual role in determining the identity and functionality of centromeres. Furthermore, *CENPA* has been demonstrated to establish the location of centromeres through the reorganization of the nucleosome structure within its folded histone core. Therefore, the transcription of *CENPA*, its stability, and deposition into chromatin is crucial for maintaining genome stability. Of note, elevated *CENPA* expression in multiple human cancer tissues, including breast, colon, and gastric, was identified by earlier studies [32,33,34], implying its nonnegligible association with cancer development. The molecular profiling of *CENPA* in human cancer was analyzed by a Oncomine analysis, which indicated that *CENPA* is highly expressed in almost 20 types of solid tumors in contrast with normal tissue [35]. *CENPA* was found to be highly overexpressed in both prostate cancer tissue and cell lines. It functions as a transcriptional regulator that modulates tumor progression and is associated with a poor outcome [36,37]. In hepatocellular carcinoma, Yongmei et al. discovered that *CENPA* depletion suppresses tumor cell growth by blocking the cell cycle and promoting apoptosis [38]. In order to ensure a proper chromosome segregation in the process of malignant transformation, cancer cells require sufficient expression of *CENPA*. However, the correlation between PDAC and *CENPA* has not been discussed yet. In our study, *CENPA* was selected specifically to build a prognostic risk model, and we also identified its principal effect for PC cells by the CERES score. Then, we found that PC patients with a higher expression of *CENPA* also experienced a shortened survival and worse prognosis, suggesting *CENPA* may work as a potential prognostic biomarker and therapeutic target. Furthermore, the relative high mRNA expression of *CENPA* in the PC cell line was found in our experiments, indicating the mechanism of *CENPA* in the tumorigenesis of pancreatic cancer worth investigation. Jeffery D et al. discovered that *CENPA* overexpression induces senescence and radiosensitivity in the presence of functional *P53* while promoting epithelial–mesenchymal transition (EMT) in tumor cells lacking functional P53 [37]. In our study, by utilizing CCK-8, wound-healing, and transwell migration experiments, we have ascertained that the downregulation of *CENPA* unequivocally hinders the proliferation and migratory potential of pancreatic cancer cells. We also identified two small molecular drugs, betulinic acid and GSK2126458, that demonstrate a robust binding affinity with the protein structure of *CENPA*, indicating they may serve as anti-PDAC drugs that also target *CENPA*. GSK2126458 (Omipalisib) is a highly selective PI3K inhibitor, and numerous studies have indicated its anti-tumor activity against various types of cancer [39,40]. Further research is warranted to explore the potential of these two drugs in the field of pancreatic cancer treatment.

In summary, our results suggest a potential relationship between hypoxia and histone lactylation in PDAC. Given that hypoxia and lactylation patterns are not yet easily applicable in clinical settings and that there is still a lack of reliable biomarkers for prognostic monitoring, we utilized 31 HALRGs to construct a preliminary prognostic assessment framework. While the risk model demonstrates some predictive potential for chemotherapy and immunotherapy, which are essential treatments for PDAC patients, further validation is required to confirm these findings. Additionally, through the integration of scRNA and mRNA expression data, we also identified *CENPA* as a potential therapeutic target for PDAC.

Nevertheless, some limitations in the study should be taken into consideration. First, real-world data from PDAC should be used to test the reliability and accuracy of our risk model. Considering that the ROC curve indicates that the predictive performance of the model is not as strong as anticipated, we may need to consider increasing the sample size to further assess the model’s predictive capabilities. Additionally, we will explore potential methods to enhance the model’s effectiveness, ensuring that it can provide more reliable prognostic assessments for PDAC patients. Next, our study only provides preliminary validation of the relationship between hypoxia- and lactylation-related genes and the prognosis of pancreatic cancer patients, as well as their association with immunotherapy. More in-depth experiments are needed to explore the specific biological mechanisms through which these genes contribute to pancreatic cancer development. Finally, further investigations are required to elucidate the complex molecular mechanisms through which the knockdown of *CENPA* suppresses pancreatic cancer proliferation and migration, especially in the context of hypoxic and high-lactate metabolic environments.

## 4. Materials and Methods

### 4.1. Data Acquisition and Processing

RNA sequencing (RNA-seq) expression data and genomic mutation data, along with corresponding clinical data of PDAC, were downloaded from The Cancer Genome Atlas (TCGA) database and Gene Expression Omnibus (GEO) database. (GSE183795, GSE62452, GSE197177, GSE78299, and GSE85916). The gene expression data of normal pancreatic tissue were obtained from the Genotype-Tissue Expression (GTEx) database (https://gtexportal.org, accessed on 12 September 2023). For subsequent analyses, the transcripts per kilobase million (TPM) values of the larger merged cohort were normalized to fragments per kilobase million (FPKM), and the “SVA” R package (version 3.50.0) was employed to correct for technical bias. The gene sets related to lactylation and hypoxia were extracted from the GSEA database.

### 4.2. The Genetic Variation Analysis of Hypoxia- and Lactylation-Related Genes with Prognostic Values

The frequency of copy number variations in the PDAC samples was obtained from the TCGA database. The “maftools” R package (version 2.18.0) was employed to visualize the mutation frequencies and oncoplot waterfalls for the genes associated with hypoxia and lactylation (HALRGs) in PDAC patients.

### 4.3. Pan-Cancer Analysis of Prognosis-Related Hypoxia and Lactylation-Related Genes

The Gene Set Cancer Analysis (GSCA) platform is an all-encompassing tool for performing Gene Set Cancer Analysis, covering the genomic, pharmacogenomic, and immunogenomic dimensions [41] ((http://bioinfo.life.hust.edu.cn/GSCA), accessed on 6 January 2024). The gene alterations of the prognosis-related hypoxia and lactylation-related genes (HALRGs), including mutation, copy number, and methylation, were conducted using the Gene Set Cancer Analysis (GSCA) database.

### 4.4. Unsupervised Clustering Analysis and Functional Enrichment Analysis

Using the expression profiles of 31 prognosis-related HLRGs, we applied the “ConsensusClusterPlus” R package (version 1.66.0) to conduct an unsupervised consensus clustering in a combined meta-cohort, which included transcriptome data from TCGA and GSE183795. To ensure the robustness of the classification, we repeated the process 50 times. The distribution of the two clusters was confirmed by using principal component analysis (PCA). To further study the biological processes and immune function differences between the two clusters, gene set variation analyses (GSVA) was employed based on the “GSVA” R package (version 1.50.0). The gene set of “c2.cp.kegg.v7.4.symbols” and “h.all.v2023.1.Hs.symbols” was obtained from the Molecular Signatures Database (MSigDB) (https://www.gsea-msigdb.org/gsea/msigdb/, accessed on 24 September 2023). The immune cell infiltration of the two clusters was studied by the single-sample gene set enrichment analysis (ssGSEA). In addition, PROGENy (Pathway RespOnsive GENes for activity inference) was used to evaluate the activation of different cell signaling pathways between the two clusters. In contrast to pathway mapping methods, PROGENy is a novel approach that addresses limitations arising from the influence of post-translational modifications and downstream signatures [42]. To compare the significantly distinct biological functions between the high- and low-risk groups, gene set enrichment analysis (GSEA) was conducted by using the “clusterProfiler” R package (version 4.10.0).

### 4.5. Construction and Validation of the Hypoxia- and Lactylation-Related Prognostic Signature

Based on the 31 genes with prognosis values selected by the univariate Cox analysis, we used multiple machine learning algorithms, including LASSO regression analysis, XGBoost, and random forest, to identify the hub genes and developed a prognostic signature. In this analysis, the meta-cohort composed of TCGA and GSE183795 combined transcriptome data was used as the training set, which was applied to establish the HALRG prognostic risk model. The risk score of each tumor sample was calculated based on the standardized expression level of the 8 hub genes and their corresponding regression coefficient. Through the median value of the risk score, tumor samples were classified into low-risk and high-risk groups. Expression data and survival information from GSE62452 (*n* = 65), GSE85916 (*n* = 80), and GSE78299 (*n* = 49) were used as three independent external test sets to validate the reliability of the risk model. In order to assess the effectiveness of the risk model, Kaplan–Meier survival curves and receiver operating characteristic (ROC) curves for 1, 3, and 5 years were generated using the “survival” (version 3.5.7) and “survivalROC” R packages (version 1.0.3.1). Additionally, univariate analysis and multivariate analysis were conducted to investigate the independent prognostic significance of the risk model.

### 4.6. Independent Prognostic Analysis and the Construction of Nomogram

To assess the independent impact of the risk score on the survival of patients with pancreatic ductal adenocarcinoma (PDAC), univariate and multivariate analyses were conducted. Subsequently, a nomogram was developed using the “regplot” R package (version 1.1), incorporating gender, grade, age, and risk score to predict survival outcomes. Additionally, a calibration curve was plotted to evaluate the accuracy of the nomogram in predicting the probabilities of 1-, 3-, and 5-year overall survival (OS).

### 4.7. Analysis of Immunologic Landscape

To enhance our comprehension of the immunologic landscape in the low- and high-risk groups, we employed algorithms, such as CIBERSORT, XCELL, TIMER, EPIC, etc., to investigate the disparities in immunocyte infiltration profiles between the two risk groups. Additionally, we utilized the “estimate” R package (version 1.0.13) to estimate the stromal levels, immunocyte infiltration degrees, and tumor purity in the PDAC samples. This analysis facilitated the exploration of immune status grouping within the risk score model. Furthermore, we conducted Spearman correlation analysis to examine the association between the risk model genes and risk score with genes related to immune escape [28], thereby gaining further insights into the immunologic characteristics of the risk score.

### 4.8. Prediction of Immunotherapy Response and Sensitivity of Chemotherapy Drugs

The Tumor Immune Dysfunction and Exclusion (TIDE) algorithm was utilized to forecast the prospective responses to an immune checkpoint blockade within the low- and high-risk groups, using tumor samples extracted from the TCGA database. To assess the chemosensitivity, we utilized “oncoPredict”, an R package (version 0.2) that predicts in vivo drug responses and identifies biomarkers for cancer patients based on cell line screening data, in order to construct translational models of drug responses in the low- and high-risk groups.

### 4.9. CERES Score and scRNA-Seq Analysis of the Signature Genes

CERES is a CRISPR-based knockout technology that looks at the specific gene dependencies of specific cancer cells and then constructs a cancer dependency map [43]. A lower CERES score suggests the gene of interest is more likely to be vital for a particular cancer cell line. The CERES score was obtained following a previous study (https://www.mdpi.com/2072-6694/13/9/2128, accessed on 24 October 2023). Classical markers of cell subtype definitions used for manual annotation were obtained from previous studies and manually annotated according to marker expression [44]. We assessed the single-cell samples by scoring them based on the expression levels of the risk genes using the “AddModuleScore” function within the “Seurat” R package (version 5.0.1). Additionally, we explored differences in the cellular communication levels between high- and low-scoring samples employing the “CellChat” R package (version 1.6.1). To investigate the protein profile of *CENPA*, the protein expression of *CENPA* in PDAC tissue was visualize in the Human Protein Atlas database (HPA) (https://www.proteinatlas.org/, accessed on 2 October 2023).

### 4.10. Drug Sensitivity Analysis and Molecular Docking

To further explore the clinical significance of *CENPA*, we conducted a correlation analysis between the expression level of *CENPA* and drug sensitivity in the CTRP and PRISM databases using the BEST website [45]. This was done to screen potential anti-pancreatic cancer drugs associated with *CENPA*. Subsequently, we selected drugs that exhibited a positive correlation with a high expression of *CENPA* in the correlation analysis and conducted molecular docking studies with them. The names, molecular weights, and 3D structures of small molecule drugs were retrieved from the PubChem database. Subsequently, we downloaded the 3D structure corresponding to the *CENPA* gene from the RCSB PDB database (http://www.rcsb.org/, accessed on 24 February 2024). AutoDock Vina software (https://vina.scripps.edu/, accessed on 25 February 2024, version 1.2.5) was employed to prepare ligands and proteins for the molecular docking process. The Affinity (kcal/mol) value was used to indicate the binding strength between the two entities. A lower Affinity (kcal/mol) value suggested a more stable binding of the ligand to the receptor. The results were visualized using Pymol (version 3.0).

### 4.11. Cell Culture, RNA Extraction, and RT-qPCR

The PDAC cell lines PANC-1, HPAF-II, CFPAC-1, CAPAN-1, ASPC-1, BXPC-3, CAPAN-2, MIA PaCa-2, and SW1990, as well as the normal pancreatic ductal epithelial cell lines hTERT-HPDE and HPNE, were acquired from the Cell Bank of the Shanghai Institute of Cells, Chinese Academy of Science, located in Shanghai, China. RNA was extracted from the cell lines using the FORGENE cell total RNA extraction kit (Cat. No. RE-03113F, FORGENE, Chengdu, China) following the manufacturer’s instructions. The primers we used in this study are as follows: β-actin forward, 5′-CATGTAC GTTGCTATCCAGGC-3′; reverse, 5′-CTCCTTAATGTCACGCACGAT-3′; *CENPA* forward, 5′-GGCGGAGACAAGGTTGGCTAAA-3′; reverse, 5′-GGCTTGCCAATTGAAGTCCACAC. β-actin served as the internal control for the mRNA analysis, and the relative expression levels of mRNA were determined using the relative quantification method. The amplification reaction consisted of the following steps: an initial denaturation at 95 °C for 1 min, followed by 39 cycles of denaturation at 95 °C for 20 s and annealing/extension at 60 °C for 1 min.

### 4.12. Western Blot

Cells were plated in six-well dishes at a density of 2 × 10^5^ cells per well, followed by subsequent digestion in a RIPA buffer supplemented with a 1% protease inhibitor cocktail (Bimake, Houston, TX, USA). The protein samples were separated using SDS-PAGE and transferred onto PVDF membranes. After blocking with 3% skimmed milk, the membranes were exposed to specific primary antibodies, including *CENPA* (ZENBIO, Chengdu, China) and β-actin (Zhongshan Golden Bridge, Beijing, China), and incubated overnight at 4 °C, followed by incubation with the secondary antibody at room temperature for 90 min. Immunoreactive bands were detected using an enhanced chemiluminescence reagent, with β-actin employed as the internal control.

### 4.13. CCK8 Assay and Clone Formation Assay

In short, cells (4 × 10^3^ per well) were plated onto 96-well plates and cultured for 0, 24, and 48 h. Subsequently, twn microliters of CCK8 reagent from TargetMol (Shanghai, China) were introduced to each well and incubated at 37 °C for 2.5 h. The absorbance was then assessed at 450 nm using a multifunctional enzyme marker. To perform clone formation assays, cells that had been transfected (2 × 10^3^ per well) were cultivated in 12-well plates for a duration of 10 to 14 days. Subsequently, the cells were immobilized through the application of a 4% formaldehyde solution in PBS and then subjected to staining with crystal violet.

### 4.14. Wound-Healing Assay and Transwell Assay

As for the wound-healing assay, cells (2.5 × 10^5^) were cultured in six-well plates until reaching full confluence; then, the culture medium was exchanged with serum-free medium, and a scratch was created by running a twenty-milliliter pipette tip across the plate, removing a line of cells. Pictures were captured at 0, 24, and 48 h under a microscope to document the progress of wound healing. For the migration assays, cells (6 × 10^4^ per well) subjected to various treatments were suspended in 200 μL of serum-free medium and added to the upper transwell cell culture chambers equipped with a polycarbonate membrane with an 8-μm pore size. Simultaneously, 800 μL of complete medium containing 10% FBS was positioned in the lower chamber to serve as a chemoattractant. Following incubation at 37 °C for 24 h, the filters were immobilized by submerging them in 4% paraformaldehyde for 15 min, followed by staining with crystal violet for 20 min.

### 4.15. Statistical Analysis

Statistical analyses in the study were performed using R software (version 4.1.2) and GraphPad Prism 9.5. Additional details regarding the statistical approaches can be found in the preceding section. Statistical significance was defined as *p* < 0.05.

## Figures and Tables

**Figure 1 ijms-25-11143-f001:**
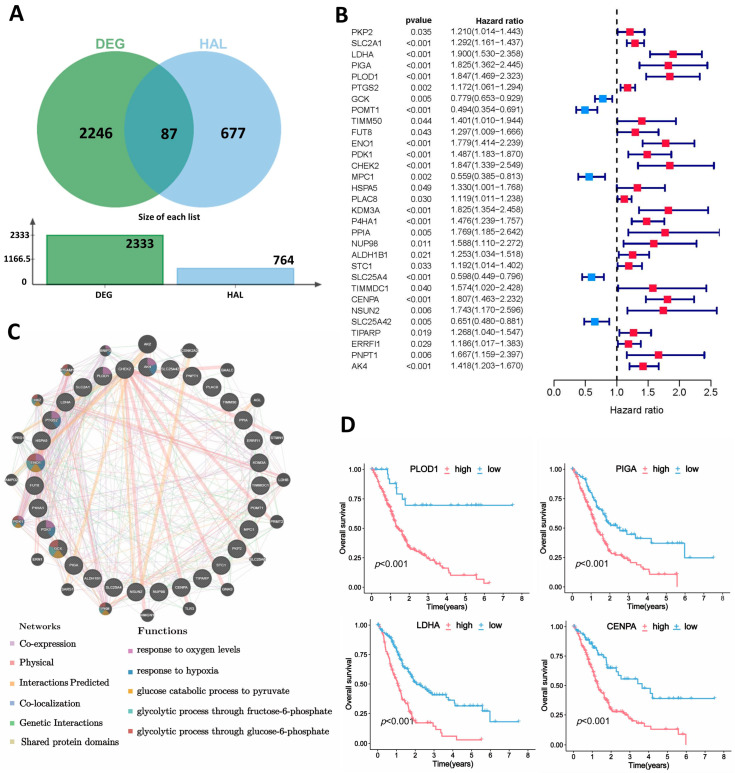
Identification of prognostic hypoxia- and lactylation-related genes (HALRGs) and mutation landscape. (**A**) Intersection of differentially expressed genes (DEGs) in PDAC samples with hypoxia- and lactylation-related genes. (**B**) Univariate Cox analysis of these genes. (**C**) Biological network integration of these prognostic genes analyzed by GeneMANIA. (**D**) Kaplan–Meier survival curve of certain prognostic genes.

**Figure 2 ijms-25-11143-f002:**
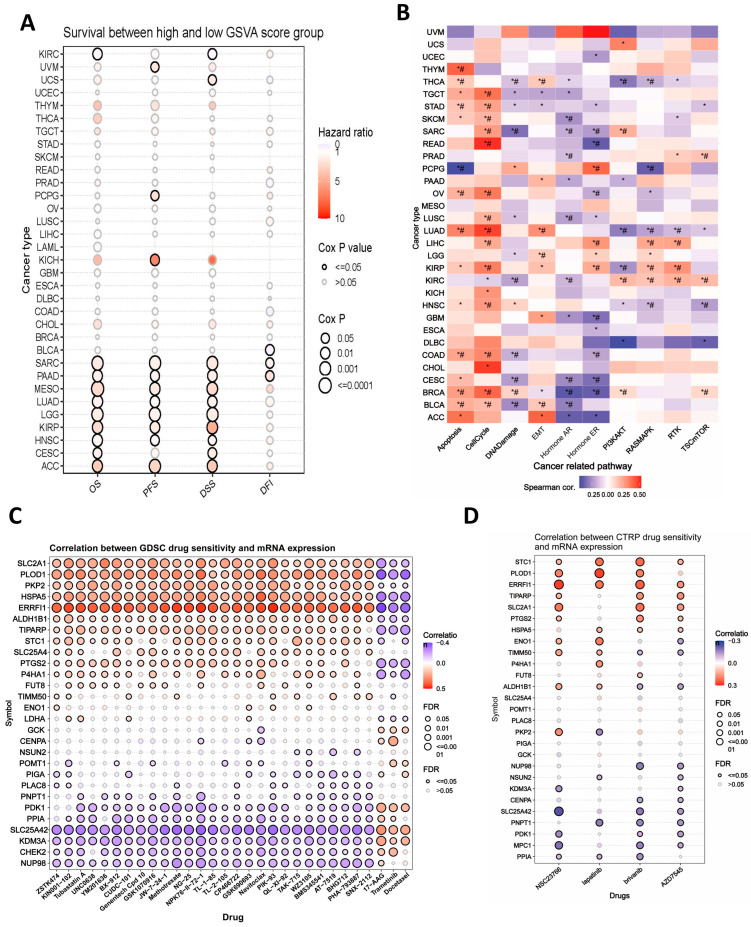
Pan-cancer analysis of the prognostic hypoxia- and lactylation-related genes. (**A**) Survival differences between high and low GSVA score groups across various cancers. (**B**) Association between GSVA scores and cancer-related pathway activity (*: *p*-value ≤ 0.05; #: FDR ≤ 0.05). (**C**,**D**) Summary of the relationship between gene expression and responsiveness of top 30 GDSC and CTRP drugs in the pan-cancer analysis.

**Figure 3 ijms-25-11143-f003:**
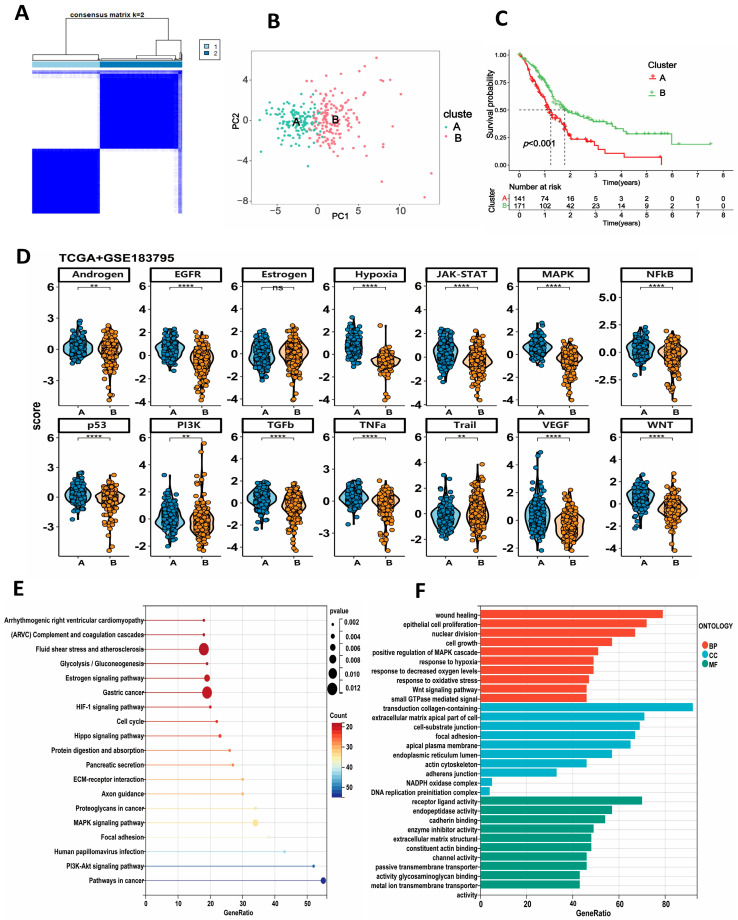
Unsupervised clustering analysis identified two PDAC subtypes with distinctive biological functional characteristics in the TCGA and GSE183795 cohorts. (**A**) Consensus matrix heatmap defining two subtypes (k = 2). (**B**) PCA indicating the significant differences in transcriptomes between the subtypes. (**C**) Survival analysis indicates cluster A has a poor prognosis compared to cluster B. (**D**) Using PROGENy (Pathway RespOnsive GENes for activity inference) to assess the pathway activation in the above two subtypes (ns *p* > 0.05; ** *p* < 0.01; **** *p* < 0.0001). (**E**) KEGG enrichment analysis of the two subtypes. (**F**) GO enrichment analysis of the two subtypes.

**Figure 4 ijms-25-11143-f004:**
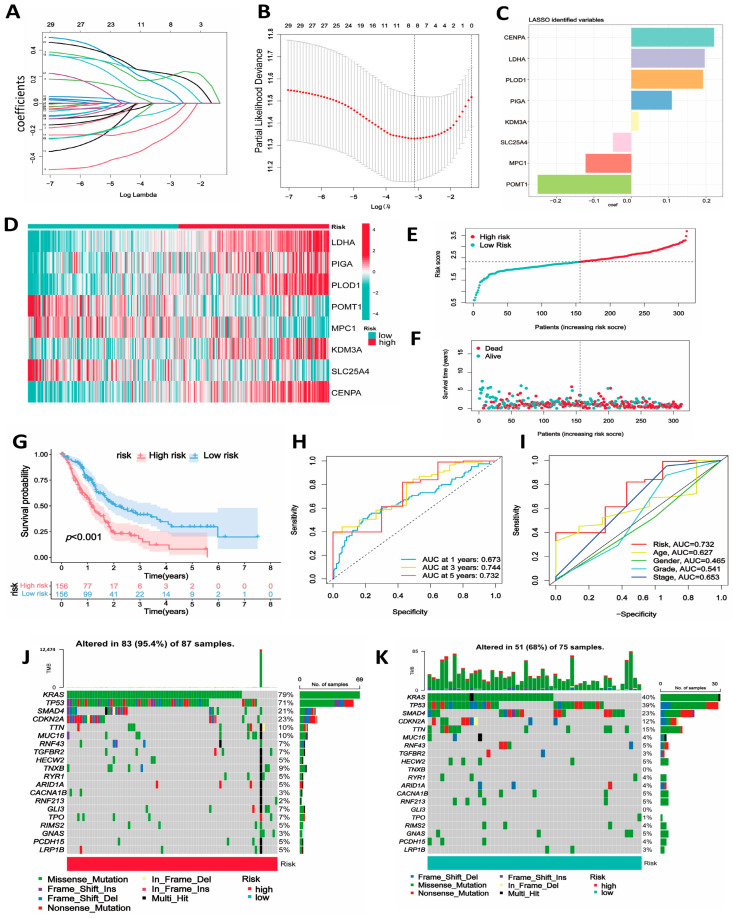
Identification of hub genes using various machine learning algorithms, and construction of a hypoxia- and lactylation-related prognostic signature for PDAC. (**A**,**B**) LASSO Cox regression was used to identify signature genes and develop a prognostic module for PDAC patients. (**C**) Bar graph of the coefficient index of the hub genes. (**D**) Heatmap of hub gene expression in the low- and high-risk groups. (**E**,**F**) Risk score distribution and survival status in the two risk groups. (**G**) Kaplan–Meier survival curve showing overall survival (OS) in the two risk groups. (**H**) ROC curves predicting the sensitivity and specificity of the risk score model for the 1-, 3-, and 5-year survival rates. (**I**) Time-dependent ROC analysis indicating the predictive power of the risk signature and other clinical characteristics. (**J**,**K**) Mutation landscape of the low- and high-risk groups.

**Figure 5 ijms-25-11143-f005:**
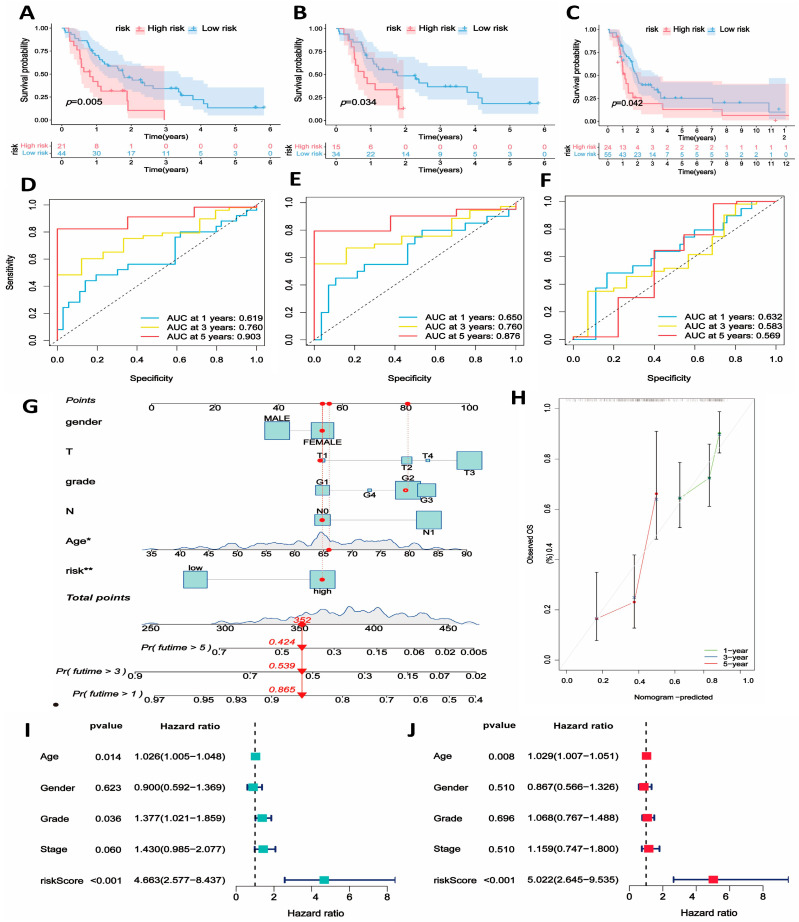
Validation of the prognostic module in independent external datasets (GSE62452, GSE78299, and GSE85916) and nomogram construction. (**A**–**C**) Kaplan–Meier analysis validating the predictive power of the prognostic model in the GSE62452, GSE78299, and GSE85916 datasets. (**D**–**F**) ROC curves demonstrating the sensitivity and specificity of the risk score model for the 1-, 3-, and 5-year survival rates in these test cohorts. (**G**,**H**) Nomogram construction integrating the risk score and clinical characteristics (* *p* < 0.05; ** *p* < 0.01). (**I**,**J**) Forest plots of the univariate and multivariate Cox regression analyses show that the risk score is an independent prognostic factor for PDAC in the training cohort.

**Figure 6 ijms-25-11143-f006:**
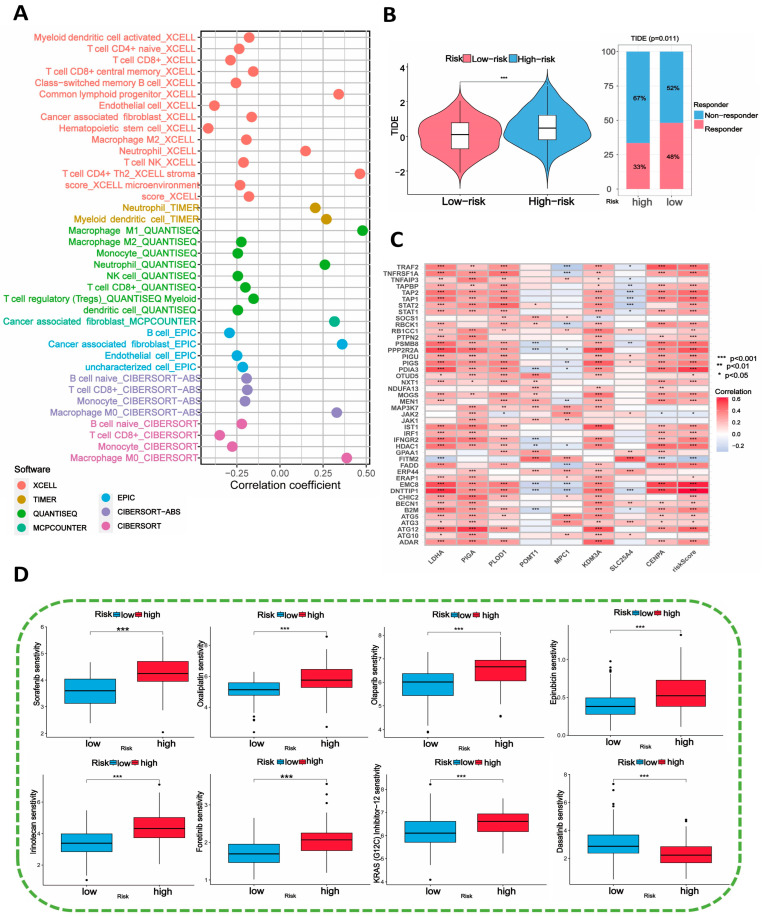
The immunogenomic landscape of signature genes and their predictive values for immunotherapy and chemotherapy. (**A**) Correlation between risk scores and immune cell abundance analyzed using various immune cell profiling methods. (**B**) Evaluation of the potential efficacy of immunotherapy in low- and high-risk groups, showing a less favorable response in the high-risk group.(*** *p* < 0.001) (**C**) Correlation analysis between signature genes and genes associated with immune evasion. (**D**) Analysis of chemotherapeutic sensitivity between the low- and high-risk groups (*** *p* < 0.001).

**Figure 7 ijms-25-11143-f007:**
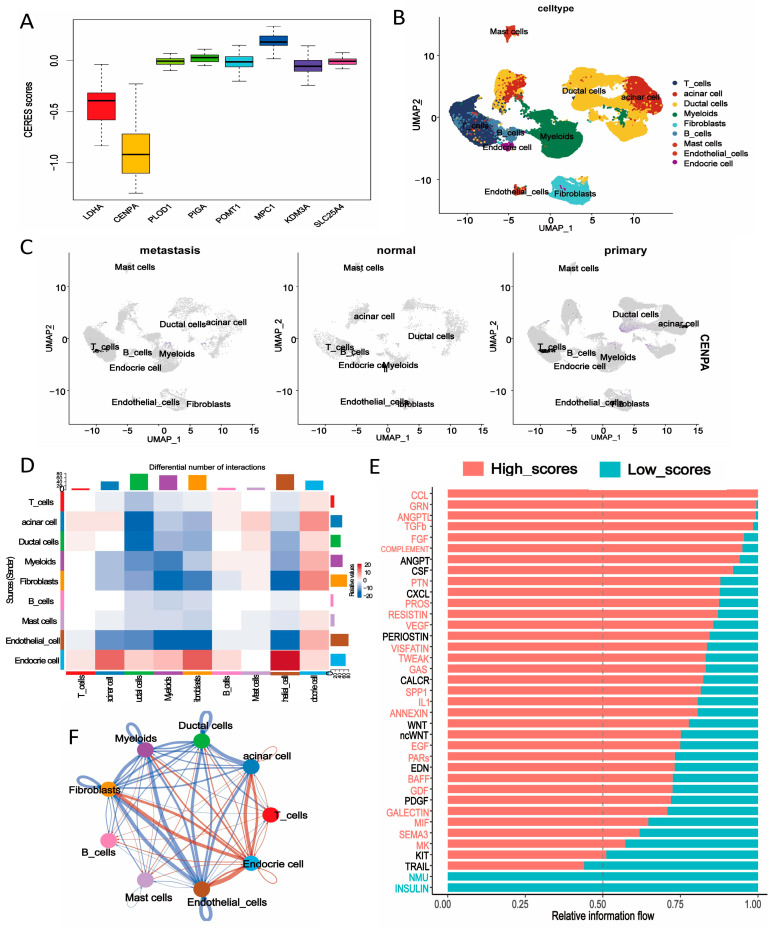
CERES score of signature genes and HAL score analysis at the single-cell level. (**A**) CERES score of signature genes. (**B**) UMAP-1 plot showing cell subtypes identified from scRNA-seq data. (**C**) Distribution of *CENPA* in metastasis, normal, and primary PADC scRNA samples. (**D**) Heatmap displaying variations in interaction numbers. (**E**) Bar graph showing key signaling pathways differing between the high- and low-scoring groups. (**F**) Circular plot visualizing differences in cell–cell communication networks between the high- and low-scoring groups.

**Figure 8 ijms-25-11143-f008:**
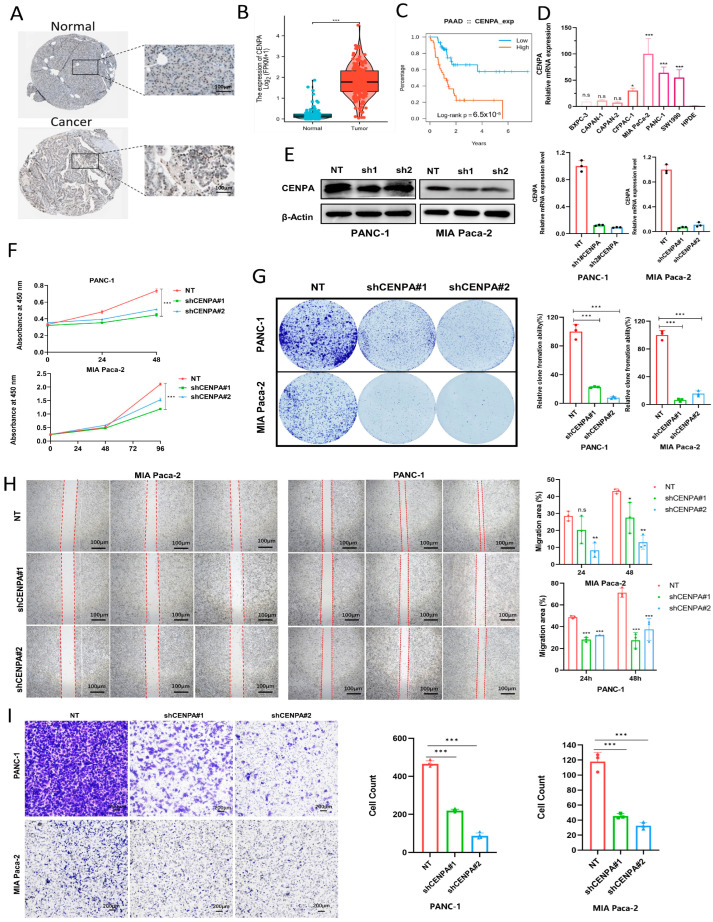
The expression profile of *CENPA* in PDAC; the knockdown of *CENPA* hampers the proliferation and migratory potential of PDAC cells. (**A**) Validation of *CENPA* expression in the HPA database. (**B**) The expression level of *CENPA* in the PDAC expression data cohort from the TCGA and GETx database. (**C**) Associations between *CENPA* expression and overall survival of PDAC patients. (**D**) Relative mRNA expression of *CENPA* in PDAC cell lines (BXPC-3, CAPAN-1, CAPAN-2, CFPAC-1, MIA PaCa-2, PANC-1, and SW1990) and HPDE normal pancreatic ductal epithelial cells. (**E**) *CENPA* knockdown in PANC-1 and MIA PaCa-2 cells verified by qRT-PCR and Western blot. The cck8 assay (**F**) and colony formation assay (**G**) show reduced cell viability in *CENPA* knockdown PANC-1 and MIA PaCa-2 cells. (**H**,**I**) Wound-healing and transwell assays indicate significantly reduced migration ability in *CENPA* knockdown PANC-1 and MIA PaCa-2 cells. *n* = 3, ns *p* > 0.05, * *p* < 0.05, ** *p* < 0.01, *** *p* < 0.001. Error bars represent mean ± SD.

**Figure 9 ijms-25-11143-f009:**
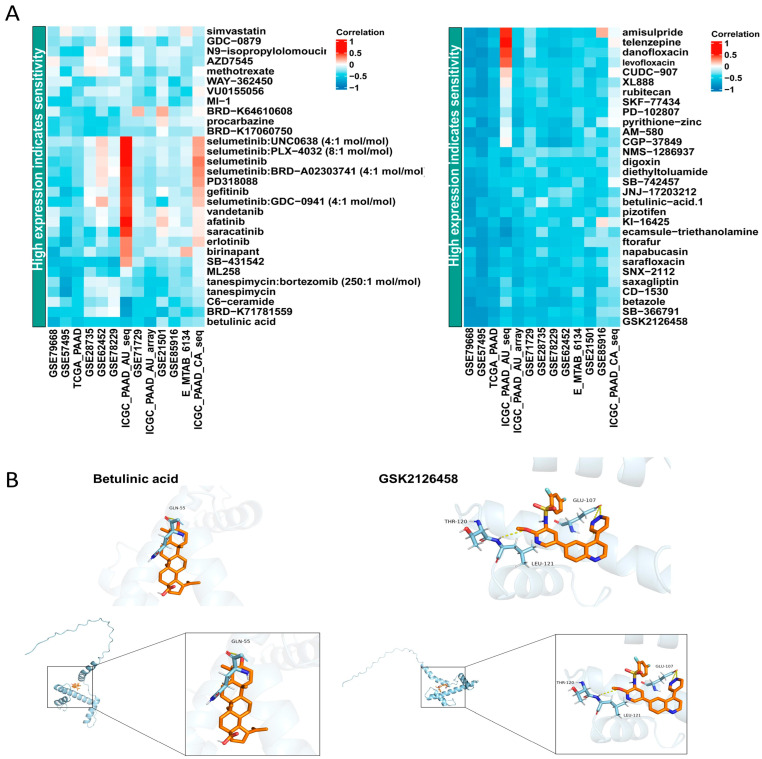
Correlation between *CENPA* expression and drug sensitivity, and molecular docking of drugs correlated with the high expression of *CENPA*. (**A**) Correlation analysis between *CENPA* expression and drug sensitivity, conducted using BEST. (**B**) Molecular docking diagrams of *CENPA* with the two drugs showing the strongest binding affinity: betulinic acid (−8.1 kcal/mol) and GSK2126458 (−8.6 kcal/mol).

## Data Availability

The original contributions presented in the study are included in the article, and further inquiries can be directed to the corresponding authors.

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
