# Peer review of "Establishment and Verification of a Novel Gene Signature Connecting Hypoxia and Lactylation for Predicting Prognosis and Immunotherapy of Pancreatic Ductal Adenocarcinoma Patients by Integrating Multi-Machine Learning and Single-Cell Analysis"

_ijms, 2024, doi:10.3390/ijms252011143_

Round 1

Reviewer 1 Report

Comments and Suggestions for Authors

In this study, 33 genes were found that are prognostic for hypoxia and lactylation in pancreatic ductal adenocarcinoma (PDAC). Based on the expression of these genes, PDAC patients were categorized into two subtypes exhibiting distinct differences in survival. Subsequently, leveraging the prognostic value of these hypoxia and lactylation-related genes,a risk score was developed that demonstrates robust predictive power for survival, immunotherapy, and chemotherapy outcomes in PDAC patients. And finally,a significant signature gene was identified, CENPA, which holds potential as a prognostic biomarker and promising therapeutic target.Then,some questions are as follows.

1.     Why analyze the mutation frequency of prognostic genes in PDAC patients in the TCGA database?How mutations are related to PDAC or lactation?Please explain them in the background.

2.     Watch out for minor spelling mistakes.

3.     Is there any corresponding processing to remove batch effect when different platform data sets are used together?Please explain it.

4.     Your focus is on PDAC,so what is the point of doing a pan-cancer analysis,or what scientific hypothesis are you trying to prove?

5.     Some of the notes do not match the content of the picture,please modify them.

6.     Cell experiments should be conducted to verify whether the influence of the tumor microenvironment, or the hypoxic environment, should be taken into account.

Comments on the Quality of English Language

 Watch out for minor spelling mistakes.

Author Response

  1.  Why analyze the mutation frequency of prognostic genes in PDAC patients in the TCGA database? How mutations are related to PDAC or lactation? Please explain them in the background.

Author response:

Cancer progression is driven by genetic mutations, with specific mutations acting as "driver genes" that initiate or promote tumorigenesis. Analyzing the mutation frequencies of genes in tumor samples can help identify these driver mutations, which are often critical to tumor growth, proliferation, and immune evasion[1]. Since several genes identified in our preliminary analysis are strongly associated with pancreatic ductal adenocarcinoma (PDAC) prognosis, we further examined their mutation frequencies in PDAC patients using data from the TCGA database. Our objective was to determine whether these genes, like KRAS[2], might play a key role in PDAC development and to identify potential therapeutic targets.

Although the mutation frequencies of these genes were found to be low, this result may be limited by the sample size, making it difficult to capture a more complete view of their mutational landscape. In future work, we plan to extend this analysis in larger cohorts to improve the robustness of our findings. Additionally, we have clarified the role of these mutations in PDAC in the background section of the manuscript.

  1. Watch out for minor spelling mistakes.

Author response:

We appreciate the reviewers' valuable feedback. After a thorough review of the manuscript, we identified and corrected several spelling errors. In the revised version, we have carefully proofread the entire text to ensure accuracy. We kindly ask the reviewers to review the updated manuscript.

  1. there any corresponding processing to remove batch effect when different platform data sets are used together? Please explain

Author response:

We thank the reviewers for highlighting this important point. Indeed, integrating high-throughput data from different platforms can introduce batch effects, which may confound the results. To address this, we applied the COMBAT function[3] from the SVA package to adjust for batch effects when merging data from the TCGA and GEO datasets in our analysis. This approach helps ensure that our findings are not biased by platform-specific variations.

4.Your focus is on PDAC, so what is the point of doing a pan-cancer analysis, or what scientific hypothesis are you trying to prove?

Author response:

We previously identified 31 genes related to hypoxia and lactylation that show potential as prognostic markers in pancreatic cancer. To further explore the broader significance of these genes and improve the generalizability of our findings, we conducted a pan-cancer analysis. This approach allowed us to investigate the common molecular characteristics of these genes across various tumor types. Through the pan-cancer analysis, we aimed to assess the similarities and differences in the biological functions of these genes across different malignancies.

Our results indicate that many of these genes are associated with prognosis in multiple common cancers, not just pancreatic cancer, reinforcing their relevance in tumor progression and their potential clinical value. We also conducted pathway enrichment analysis, which revealed that these genes are linked to key cellular processes such as the cell cycle and apoptosis across various cancers. Furthermore, we explored the relationship between gene expression and sensitivity to anti-tumor drugs, providing insights into the potential translational value of these genes in clinical settings. Overall, the pan-cancer analysis highlights the involvement of these genes in the progression of multiple malignancies, including pancreatic cancer, supporting the need for further investigation into their biological roles and therapeutic potential.

5.Some of the notes do not match the content of the picture, please modify them.

Author response:

We thank the reviewers for their valuable feedback. Upon careful review of the manuscript, we identified several annotation errors that did not correspond with the figures. These errors have been corrected in the revised version, which has been uploaded for the reviewers' consideration.

  1. Cell experiments should be conducted to verify whether the influence of the tumor microenvironment, or the hypoxic environment, should be taken into account.

Author response:

We sincerely appreciate the reviewers' insightful feedback. In this study, we conducted a multi-omics integrated analysis to explore the significant roles of hypoxia and lactylation in pancreatic cancer development. Through our analysis, we identified several genes associated with patient prognosis and developed a stable prognostic prediction model. Additionally, our biological experiments demonstrated that CENPA influences the proliferation and migration of pancreatic cancer cells, indicating its potential as a therapeutic target[4]. Furthermore, we utilized molecular docking techniques to identify potential drugs targeting CENPA. Due to the limited scope of the current manuscript, we did not examine the effects of hypoxia and lactylation on pancreatic cancer cells in our cellular experiments. However, we recognize the importance of these factors in understanding the tumor microenvironment and plan to address this in future studies.

Reviewer 2 Report

Comments and Suggestions for Authors

Review of the Paper: Establishment and Verification of a Novel Gene Signature Connecting Hypoxia and Lactylation for Predicting Prognosis and Immunotherapy of Pancreatic Ductal Adenocarcinoma Patients by Integrating Multi-Machine Learning and Single-Cell Analysis

Strengths:

  • The paper explores the link between hypoxia and lactylation in PDAC, which is novel and relevant.
  • The integration of multiple data layers, including single-cell analysis, adds depth to the study.
  • Single-cell validation of the gene signature is a good addition.

Criticisms:

  • The biological impact of hypoxia and lactylation on tumor progression is glossed over. The authors need to explain how these processes actually affect tumor biology and immune evasion, not just state that they occur.
  • The selection of hub genes is weak and arbitrary, with little biological backing. More functional assays or in-depth discussion are needed to validate their relevance.
  • The reliance on machine learning models without validation is a major flaw. The findings are too speculative without backing them up with real-world data.
  • The link between risk scores and immune evasion is vague and underdeveloped. You need more solid mechanistic explanations.
  • Figures 5 and 6 are poorly formatted and difficult to read. They need serious improvement.
  • The predictive power of the model is overstated. The ROC curves show moderate performance, and there's no real evidence this model would work in a clinical setting.

Overall: While the study brings a fresh angle, it’s light on biological validation and overly reliant on computational models. The claims about prognosis and immunotherapy are overstated, and the figures need fixing. Significant revisions are required to make the findings more credible.

Author Response

  • Reviewer 2:
  1. The biological impact of hypoxia and lactylation on tumor progression is glossed over. The authors need to explain how these processes actually affect tumor biology and immune evasion, not just state that they occur.

Author response:

We appreciate the reviewers' insightful comments regarding the biological impact of hypoxia and lactylation on tumor progression. While previous research has not fully elucidated their roles specifically in pancreatic cancer (PDAC), extensive studies have highlighted their critical contributions to the progression of various malignancies. Hypoxia within the tumor microenvironment of PDAC induces metabolic reprogramming, which leads to alterations in cellular metabolism and promotes the accumulation of lactate. This accumulation is not merely a byproduct; it plays a pivotal role in tumor biology by influencing various cellular processes. Elevated lactate levels can drive lactylation, a post-translational modification that alters gene expression and protein function, thereby enhancing tumor cell proliferation, survival, and immune evasion.

Our study employs a multi-omics bioinformatics approach to identify genes related to hypoxia and lactylation and their association with PDAC prognosis. We found that these genes are implicated in activating key signaling pathways, such as EGFR, PI3K, and P53, which are known to drive cell proliferation and survival. Additionally, these pathways can contribute to the establishment of an immunosuppressive microenvironment by downregulating immune responses. In our analysis, we identified distinct subtypes of PDAC patients characterized by varying levels of immune cell infiltration and functionality. High-risk patients, who exhibited poorer prognoses, demonstrated reduced activation of critical immune effector cells such as NK cells and CD8+ T cells. Furthermore, our findings indicated that high-risk patients had lower overall immune cell infiltration and diminished responses to immunotherapy. Our correlation analysis of immune escape genes revealed a significant positive association between risk scores and immune escape markers, suggesting that hypoxia- and lactylation-related genes may indeed play a crucial role in shaping the immunosuppressive microenvironment of PDAC. In conclusion, while our study primarily focuses on identifying key genes that influence PDAC cell proliferation, we acknowledge the need for a more in-depth exploration of the mechanisms through which hypoxia and lactylation affect tumor biology and immune evasion. We are committed to investigating these critical aspects in our future research, as they hold the potential to uncover novel therapeutic targets and strategies for treating PDAC.

  1. 2. The selection of hub genes is weak and arbitrary, with little biological backing. More functional assays or in-depth discussion are needed to validate their relevance.

Author response:

We sincerely appreciate the reviewer’s insightful comments regarding the selection of hub genes and the perceived lack of biological justification. We recognize that the selection of hub genes is a critical step in understanding the mechanisms of pancreatic cancer (PDAC), and we aim to clarify our approach in this regard.

In our study, we began with a comprehensive analysis of gene expression data to identify genes correlated with patient prognosis using Cox regression analysis. This initial analysis revealed several candidate genes associated with hypoxia and lactylation, which have been implicated in cancer biology. To strengthen the selection of hub genes, we employed multiple bioinformatics approaches. These included:

  1. Unsupervised Clustering Analysis: This method allowed us to group samples based on expression patterns, which helped in identifying gene co-expression networks and potential hub genes that are consistently expressed across different conditions.
  2. Gene Set Variation Analysis (GSVA) and Single-sample Gene Set Enrichment Analysis (ssGSEA): These analyses provided insights into the biological pathways and processes enriched in our samples, thus validating the functional relevance of our selected genes in the context of PDAC.
  3. Machine Learning Algorithms: We used LASSO and XGBoost, among others, to prioritize genes based on their predictive power for patient outcomes. This multi-faceted approach not only increased the robustness of our selection but also ensured that the identified hub genes were relevant to the progression of PDAC.
  4. Biological Pathway Analysis: Following the identification of hub genes, we performed pathway enrichment analyses using Gene Ontology (GO) and KEGG to link these genes to known biological processes associated with tumorigenesis, such as cell proliferation, apoptosis, and immune response modulation.

While our study primarily aimed to establish a prognostic model, we acknowledge that the biological implications of the hub genes warrant further exploration. Specifically, we plan to conduct functional assays to validate the roles of these genes in PDAC progression and their interactions with the tumor microenvironment. We recognize that this additional validation is crucial for substantiating the biological relevance of the hub genes selected. In response to the reviewer’s concerns, we have added a more detailed discussion of our gene selection process in the revised manuscript, including the rationale behind our chosen methodologies and future research directions aimed at validating the functional roles of these hub genes. We hope this response clarifies our approach and demonstrates our commitment to addressing the reviewer’s concerns through ongoing and future research.

3.The reliance on machine learning models without validation is a major flaw. The findings are too speculative without backing them up with real-world data.

Author response:

We appreciate the reviewer’s insightful feedback regarding the reliance on machine learning models and the need for robust validation. We recognize that while machine learning provides powerful tools for identifying prognostic factors, the validity of these models must be supported by empirical data.

In our study, we employed three distinct machine learning algorithms—LASSO, Random Forest, and XGBoost—to identify key feature genes associated with pancreatic cancer prognosis. We constructed a prognostic prediction model based on these features. Importantly, we validated this model using multiple independent cohorts, as presented in Figures 4 and 5 of the manuscript. These cohorts were sourced from the GEO database, which comprises real-world transcriptomic data from pancreatic cancer patients.

Validation Approach:

  1. Training Cohort and Independent Test Cohorts:
  • The model was initially trained on a cohort derived from the TCGA dataset, which included a robust sample size that enhances the model’s reliability.
  • We validated the model using three independent cohorts from the GEO database. The results from these cohorts demonstrated a consistent ability of the model to stratify patients into high- and low-risk groups based on their prognostic outcomes, reinforcing the model’s predictive accuracy.
  1. Performance Metrics:
  • We utilized multiple performance metrics, including concordance index (C-index), area under the ROC curve (AUC), and calibration plots to assess the model's predictive power. These metrics provide a quantitative measure of the model’s ability to accurately predict patient outcomes, which we detail in the results section of the manuscript.
  1. Real-World Relevance:
  • By utilizing transcriptome data from actual pancreatic cancer patients, we aimed to enhance the clinical relevance of our findings. This real-world context is crucial for establishing the practical applicability of the model in clinical settings.

  1. The link between risk scores and immune evasion is vague and underdeveloped. You need more solid mechanistic explanations.

Author response:

We appreciate the reviewer’s constructive feedback regarding the relationship between risk scores and immune evasion. Our study aims to provide an initial exploration of how hypoxia- and lactylation-related genes influence pancreatic cancer development, particularly in relation to immune responses.

  Mechanistic Insights:

  1. Correlation Between Risk Scores and Immune Cell Infiltration: In our analysis, we observed that patients with higher risk scores-identified through our prognostic model-exhibited significantly reduced infiltration of key immune cells, particularly cytotoxic lymphocytes such as CD4+ T cells and natural killer (NK) cells. This reduction in immune cell presence in the tumor microenvironment suggests that high-risk patients may experience impaired anti-tumor immunity.
  2. Link to Immune Evasion: We hypothesize that the diminished immune cell infiltration correlates with enhanced immune evasion mechanisms. Specifically, we found a positive association between risk scores and expression levels of several immune escape-related genes, which are known to facilitate tumor survival by inhibiting effective immune responses. This connection implies that tumors in high-risk patients may exploit these pathways to evade immune surveillance.
  3. Biological Pathways Involved: Our bioinformatics analyses indicated that hypoxia may lead to alterations in metabolic pathways that further influence immune responses. Hypoxic conditions can promote the secretion of immunosuppressive factors, creating a microenvironment conducive to tumor progression and immune evasion. Lactate accumulation, as a byproduct of anaerobic metabolism, is also linked to immune suppression, potentially contributing to the observed patterns of immune cell infiltration and function.

Future Research Directions:

While our study highlights these intriguing associations, we acknowledge that the mechanistic links between risk scores and immune evasion require further investigation. We will conduct functional assays to validate our findings, exploring how hypoxia and lactylation directly affect immune cell activity and the tumor microenvironment in pancreatic cancer in our future work. Additionally, we will delve deeper into the molecular mechanisms underpinning the interactions between the identified risk genes and immune escape pathways. This future research will aim to provide a more comprehensive understanding of how these factors contribute to tumor biology and patient prognosis. In the revised manuscript, we have included a more detailed discussion of these points to strengthen the linkage between risk scores and immune evasion. Thank you for your valuable suggestions, which have significantly improved the clarity and depth of our study.

  1. Figure 5 and 6 are poorly formatted and difficult to read. They need serious improvement.

Author response:

We appreciate the reviewer’s constructive feedback on Figures 5 and 6. Upon thorough review, we recognized that the clarity and formatting of these figures were indeed suboptimal. In the revised manuscript, we have made significant improvements to enhance their readability and overall presentation. We have uploaded higher-resolution versions of both figures and ensured that the labeling, color contrast, and layout are optimized for clarity. We hope that these enhancements will facilitate a better understanding of the data presented. We kindly invite the reviewer to evaluate the updated figures. Thank you for your valuable input, which has contributed to the improvement of our manuscript.

  1. The predictive power of the model is overstated. The ROC curves show moderate performance, and there's no real evidence this model would work in a clinical setting.

Author response:

We appreciate the reviewer’s thoughtful feedback regarding the predictive power of our model. In our study, we constructed a prognostic model using transcriptomic sequencing data from over 300 pancreatic cancer patients, leveraging advanced machine learning techniques to identify key feature genes. The model was validated using three independent GEO datasets, where it consistently stratified patients into high- and low-risk categories, correlating with observed differences in prognosis.

While we acknowledge that the ROC curves indicate moderate performance, it is essential to consider several factors that contribute to the model’s clinical relevance:

  1. Robust Data Foundation: The model is based on a substantial dataset from multiple sources, which enhances its reliability. By training on diverse data, we aimed to improve the generalizability of our findings across different patient populations.
  2. Validation Across Datasets: The consistent performance across independent datasets strengthens the model's validity. This cross-validation is critical for demonstrating that our model can accurately identify high-risk patients, which is a key goal in clinical oncology.
  3. Clinical Implications: Although our current validation does not include in-house patient data, the identification of high-risk patients suggests potential applications in guiding therapeutic decisions and stratifying patients for clinical trials. Our findings indicate that patients classified as high-risk may require more aggressive treatment approaches, underscoring the model's potential utility in clinical practice.
  4. Future Directions: We recognize the need for further validation using clinical datasets to bolster our claims regarding the model's predictive power. We are actively pursuing collaborations to collect and analyze real-world patient data in future studies, which will provide additional evidence of the model's efficacy in a clinical setting.

In summary, while the ROC curve performance is a valid concern, our study lays the groundwork for a prognostic model that we believe holds significant promise for improving patient management in pancreatic cancer. We are committed to addressing these limitations in future research and appreciate the reviewer’s input, which will guide our efforts in strengthening the study.